# BENCHMARKING VISUAL FAST MAPPING: PROBING VLMS' TEST-TIME IMAGE-TEXT ALIGNMENT

## ABSTRACT

Visual Fast Mapping (VFM) describes the human ability to rapidly formulate novel visual concepts from minimal examples by drawing upon prior experience and knowledge. This capability is a cornerstone of inductive reasoning and has been extensively studied in cognitive science. In the realm of computer vision, early attempts sought to replicate this capability through one-shot learning methods but achieved only limited generalization. Despite recent advancements in Visual Language Models (VLMs), which are trained on large-scale image-text corpora, this human-like ability remains elusive. In this paper, we introduce *VFM Bench*, a benchmark specifically designed to evaluate VFM capabilities in realistic industrial scenarios. Our evaluation reveals a significant performance gap of over $19.0\%$ between human proficiency and current VLMs. We observe that most VLMs tend to rely purely on visual discriminative features rather than leveraging their ingrained language knowledge for test-time alignment. Notably, while emerging visual reasoning models demonstrate promising initial improvements, a substantial gap compared to average human performance persists. This suggests a promising direction towards more effectively leveraging cross-modal information in context. The code and dataset for *VFM Bench* are anonymously available at: `https://anonymous.4open.science/r/VisualFastMappingBenchmark`.

## 1 INTRODUCTION

Visual Fast Mapping (VFM), the human ability to form new visual concepts from only a few examples, is a cornerstone of research in cognitive science and developmental psychology Carey et al. (1978); Carey and Bartlett (1978); Landau et al. (1988); Weismer et al. (1999); Alt (2013); Lieberman et al. (2022). Studies have shown that even infants can rapidly perform this mapping by aligning cross-modal information, exploiting both intra-class similarities and inter-class differences Imai et al. (1994); Yee et al. (2012). As children mature, this process becomes increasingly efficient by leveraging prior knowledge, a hallmark of inductive reasoning Markman (1989); Klein et al. (2008); Suffill et al. (2022), as illustrated in Figure 1. *This raises a pivotal question: Can modern Vision-Language Models (VLMs) approximate this human-level VFM by leveraging the vast knowledge inherent in their pre-trained models for novel image-text alignment at test time?*

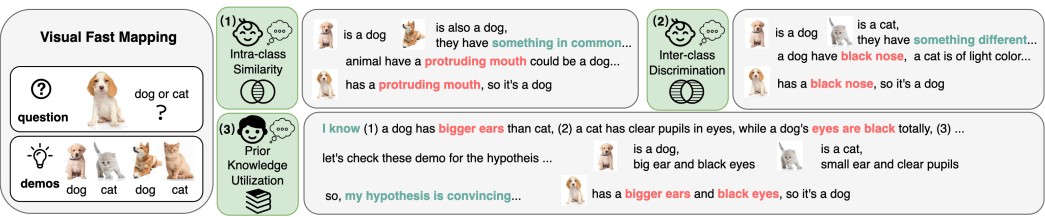

Figure 1: **Three Strategies of Visual Fast Mapping in human children.** Green words highlight key points of each strategy, while red words denote the anchor features binding to the concept.

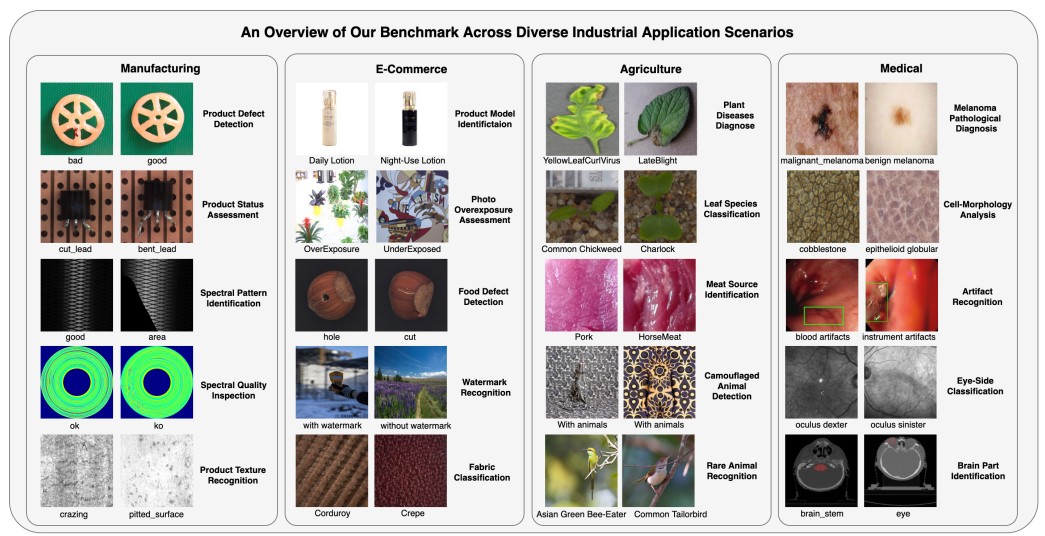

Figure 2: **A glimpse of our *VFM Bench*,** comprising 4200 query images of 512 concepts spanning 171 tasks, collected from 31 open-source datasets. These tasks present visual concepts that are novel to most individuals yet can be easily learned from a few examples by leveraging common knowledge.

Image classification has historically been the cornerstone of computer vision. Deep convolutional neural networks have achieved significant success in industrial applications by learning from extensive domain-specific datasets. However, the high cost of data collection and annotation underscores the need for human-level VFM. Early researchers pursued one-shot learning methods Fei-Fei et al. (2006); Lake et al. (2011); Vinyals et al. (2016), yet these approaches exhibited limited generalization beyond the narrow distribution of their support examples and therefore failed to deliver substantial improvements in real-world settings. In recent years, the field has shifted toward VLMs trained on interleaved image-text corpora, which can follow multi-modal instructions and leverage sufficient language knowledge. VLMs are expected to exhibit greater generalization and adaptability compared to traditional computer vision models.

Despite this promise, research on the VFM ability of VLMs is still in its infancy. On the one hand, pioneering works on visual in-context learning suggest that VLMs only imitate the answering style of example text Chen et al. (2024a); Jiang et al. (2024); Li (2025). However, these conclusions are not applicable to our situation due to an over-reliance on language generation ability rather than visual concept understanding. On the other hand, researchers have created visual inductive reasoning benchmarks inspired by human IQ tests Barrett et al. (2018); Zhang et al. (2019); Nie et al. (2021). However, none of these benchmarks truly incorporate real-world visual concept learning tasks. In summary, current vision benchmarks either focus on language generation tasks or on understanding abstract puzzles. Neither fully tests a model's capacity for VFM despite its importance.

In this paper, we introduce a Visual Fast Mapping Benchmark (*VFM Bench*), drawn from vertical domains as shown in Figure 2. Due to a small percentage of related training data, even state-of-the-art models perform worse than a random policy. Thus, in a few-shot setting, they can only rely on the provided examples, mirroring how humans learn new visual concepts. In our experiments, crowd-sourced human participants achieve an impressive improvement in accuracy, whereas most VLMs struggle. We then delve into the underlying mechanisms and identify that inter-class discrimination still plays a predominant role with limited integration of prior language ability, revealing a disparity between VLMs and human intelligence.

The contributions of this work are summarized as follows: (1) We highlight the VFM ability as a key component of visual reasoning, a concept grounded in cognitive science insights into human concept learning and long-sought by vision system researchers. (2) We propose *VFM Bench*, featuring industry-inspired classification tasks, and report comprehensive experiments comparing state-of-the-art (SOTA) VLMs with human performance, revealing a pronounced gap. (3) We identify a feasible path to bridge this gap by attributing the models' limitations to the insufficient utilization of prior linguistic knowledge for test-time cross-modal alignment. This suggests that current visual reasoning models exhibit early-stage learning patterns analogous to human cognitive processes. We believe

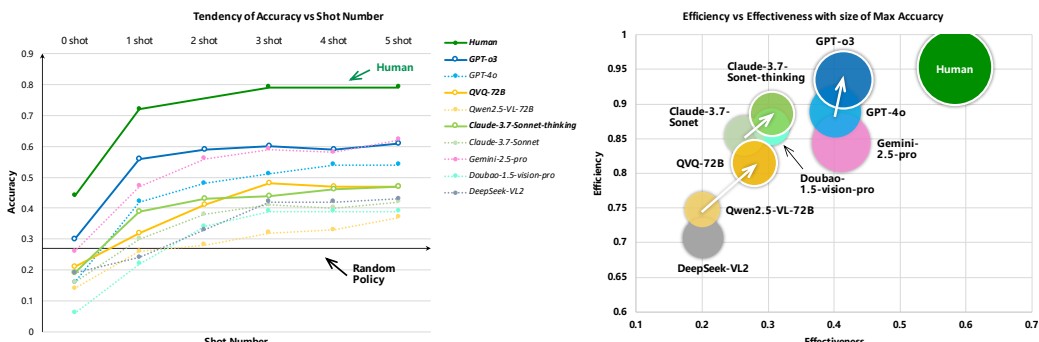

Figure 3: **The results on *VFM Bench*** reveal a significant performance gap in VFM between current VLMs and humans. Even the state-of-the-art (SOTA) model, GPT-4o, exhibits a deficit of $19.0\%$ in maximum accuracy and $16.9\%$ in effectiveness compared to average human performance. Moreover, visual reasoning models show notable improvements over their traditional counterparts, achieving average gains of $4.8\%$ in efficiency and $4.4\%$ in effectiveness.

these results offer novel insights into contemporary VLMs and pave the way for the development of more advanced vision systems for industrial applications and future embodied intelligence that are capable of handling the diverse and novel visual information encountered in real-world scenarios.

## 2 RELATED WORK

**Analysis of Visual In-Context Learning** The development of VLMs has spurred research interest in their capacity for visual in-context learning (VICL). Flamingo Alayrac et al. (2022) first demonstrated that VLMs could perform few-shot learning on visual tasks, a capability inherited from the base language model. Building upon this, subsequent research has focused on exploring its underlying mechanism. Chen et al. (2023) and Shukor et al. (2024) highlighted that the benefits of VICL are mainly driven by the text in the demonstration examples, while the visual information seems to have little impact. Moreover, Yang et al. (2024) and Doveh et al. (2024) observed the instability of VICL, indicating that more demos might degrade performance in some circumstances, while Jiang et al. (2024) reported that a large number of demos can significantly improve model performance. In summary, current analyses of VICL remain inconclusive and the phenomenon has not been systematically evaluated, despite its critical importance in intelligent emergence.

**Visual In-Context Learning Benchmark** Li et al. (2023) and Zhao et al. (2024a) both constructed multi-modal in-context instruction datasets to enhance complex instruction-following and empower perception, reasoning, and planning abilities. Zong et al. (2025) has established a benchmark for VICL that eliminates linguistic influence, yet it mainly focuses on the meta-learning field (learning a task's instruction from examples) rather than VFM. Similar to our goal, Tai et al. (2024) and Zhao et al. (2024b) have aimed to evaluate whether VLMs can learn new visual concepts in context; however, the former is detached from reality by using virtual unseen objects from image generation models, while the latter only focuses on spatial concepts.

**Visual Reasoning Benchmarks** Several visual reasoning benchmarks have recently been proposed, in which VFMs also play a crucial role. Jiang et al. (2023) and Wu et al. (2025) curated Bongard-style problems based on human IQ tests, while Nie et al. (2021) and Zhang et al. (2019) collected abstract reasoning tasks featuring geometric or symbolic patterns. Additionally, benchmarks such as Teney et al. (2019) and Bitton et al. (2022) focus on analogical reasoning over image pairs, requiring models to infer relationships between visual inputs. However, these benchmarks often lack grounding in real-world applications and do not explicitly target the core process of entity concept formation, which is fundamental to human visual cognition.

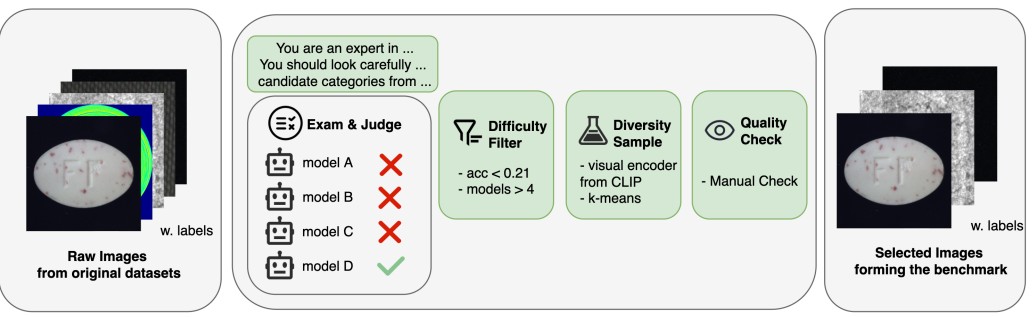

Figure 4: **A case of the defined problem**, including an instruction $\mathcal{I}$, a supporting set $\mathcal{S}$, consisting of image-text pairs for each potential category, and a query image $x_q$, the model is asked to response $y_q$ as the most possible category.

Figure 5: **The overview of raw data collection pipeline**. Difficulty filter employs five mainstream VLMs as voters to discard easy cases. The diversity sampling stage utilized k-means clustering to maximize diversity. Finally, a quality check is conducted manually.

## 3 FRAMEWORK

### 3.1 PROBLEM SETTING

The questions in *VFM Bench* can be viewed as visual inductive reasoning problems, requiring the model $F_\theta$ to estimate the label $\hat{y} \in \mathcal{Y} = \{1, \ldots, n\}$ from a query image $x_q$, based on the text instruction $\mathcal{I}$ and the support set $\mathcal{S}_k^n = \left\{(x_k, y_k)\right\}_k^n, y_k \in \mathcal{Y}$, where $n$ is the number of candidate categories and $k$ denotes the number of support examples for each category. We ensure that the query image-label pair $(x_q, y_q) \in \mathcal{D}_{\text{query}}$ and the support image-label pairs $(x_k, y_k) \in \mathcal{D}_{\text{demo}}$ are both drawn from the same downstream task using a random sampling strategy, denoted as $\mathcal{D} = (\mathcal{D}_{\text{demo}}, \mathcal{D}_{\text{query}})$. A $n$-way-$k$-shot classification problem is defined as follows:

$$F_\theta(\mathcal{I}, S_k^n, x_q) = \hat{y} = \arg\max_{c \in \mathcal{Y}} p_\theta(c \mid \mathcal{I}, \mathcal{S}_k^n, x_q). \tag{1}$$

The problem degrades to a zero-shot prediction when the supporting set $\mathcal{S}_k^n$ does not exist.

### 3.2 BENCHMARK CONSTRUCTION

In recent years, numerous high-quality datasets for perception or classification tasks in various domains have been established. Our benchmark primarily focuses on four significant industries: Agriculture, Manufacturing, Medicine, and E-Commerce, where tasks require specialized domain knowledge. More than 30,000 concept images from 31 datasets have been collected as the raw data, further details of which are provided in the Appendix.

As illustrated in Figure 5, we employed a three-stage pipeline to curate query images from the raw data, ensuring the benchmark's difficulty, diversity, and quality. First, a difficulty filter was applied to exclude samples deemed insufficiently challenging, using five mainstream mod-

els as judges ( Qwen2.5-VL-72B Bai et al. (2025), Doubao-1.5-vision-pro-32k-250115 ByteDance (2025), DeepSeek-VL2 DeepSeek Team (2024), GPT-4o OpenAI (2024) and Gemini-2.5-pro-exp-03-25 DeepMind (2025)). An image's "difficulty score" was calculated as the average of these individual model scores. Images that received an average score below a predefined threshold and were evaluated by at least a specified minimum number of models were identified as "difficult" candidates.

Subsequently, to promote diversity, we employed a CLIP visual encoder to extract image features, followed by k-means clustering to sample representative images from each industry. Finally, a manual review ensured the clarity and answerability of the selected queries. This entire process yielded a collection of 4,200 high-quality, diverse, and appropriately challenging images in *VFM Bench*, covering 512 concepts across 171 tasks, as demonstrated in Table 8 and Figure 9.

After the collection of query images, the data must be reorganized to suit the problem definition for VLM comprehension. For a single inference process, $\mathcal{I}$, $\mathcal{S}$, $\mathcal{X}$, and $\mathcal{Y}$ are required, as specified in Equation 1. The basic formats of $\mathcal{I}$, $\mathcal{X}$, and $\mathcal{Y}$ have been introduced in Figure 4. In $k$-shot settings, $k$ images are randomly selected from each candidate category to constitute $\mathcal{S}$. Moreover, different variants in the formatting and content of $\mathcal{I}$ and $\mathcal{S}$ are discussed in the Experiments Section to support the mechanism analysis experiments.

### 3.3 METRICS

The basic metric is the absolute accuracy $\text{Acc}_k^n(\theta, \mathcal{D}) = \frac{1}{M} \sum_{m=1}^{M} \mathbf{1}\{\hat{y}_q^{(m)} = y_q^{(m)}\}$, measured by the exact match strategy. Based on accuracy, we also consider other statistical indicators to measure different aspects of a VFM's ability, inspired by the discipline of human cognition.

The first aspect is efficiency, denoting how fast a model can capture the anchor visual features to form a concept with minimal examples. Inspired by Zong et al. (2025), the metric $\eta(\theta, K, \mathcal{D})$ is defined.

$$\eta(\theta, K, \mathcal{D}) = \frac{\sum_{k=1}^{K} [\text{Acc}_k(\theta, \mathcal{D}) - \text{Acc}_0(\theta, \mathcal{D})]}{K(\max_{1 \le k \le K} \text{Acc}_k(\theta, \mathcal{D}) - \text{Acc}_0(\theta, \mathcal{D}))}. \tag{2}$$

Another aspect to consider is the effective benefit gained from examples. To evaluate this capability, we introduce $\delta(\theta, K, \mathcal{D})$, which measures the maximum performance increase due to the examples provided.

$$\delta(\theta, K, \mathcal{D}) = \frac{\sum_{k=1}^{K} [\text{Acc}_k(\theta, \mathcal{D}) - \text{Acc}_0(\theta, \mathcal{D})]}{\sum_{k=1}^{K} [1 - \text{Acc}_0(\theta, \mathcal{D})]}. \tag{3}$$

These two metrics both lie within the range $[0, 1]$. Specifically, $\eta$ quantifies the average ratio of the actual benefit to the maximum achieved benefit, whereas $\delta$ measures the average ratio of the actual benefit to the total potential improvement space.

Moreover, aiming for a deeper analysis in the following sections, we denote $\mathcal{F}(\mathcal{D})$ as a data processing method applied to the dataset, and introduce $\phi(\mathcal{F}, \theta, K, \mathcal{D})$ to represent the performance impact of an ablation, where $K$ denotes the maximum number of support examples, expressed as:

$$\phi(\mathcal{F}, \theta, K, \mathcal{D}) = \frac{\sum_{k=0}^{K} [\text{Acc}_k(\theta, \mathcal{F}(\mathcal{D})) - \text{Acc}_k(\theta, \mathcal{D})]}{\sum_{k=0}^{K} \text{Acc}_k(\theta, \mathcal{D})}. \tag{4}$$

## 4 EXPERIMENT

### 4.1 EXPERIMENT SETTING

**Models.** We evaluate the following mainstream models: **(1) Visual Language Models**: Claude-3.7-Sonnet-20250219 Anthropic (2025), Doubao-1.5-vision-pro-32k-250115 ByteDance (2025), Qwen2.5-VL-72B Bai et al. (2025), DeepSeek-VL2 DeepSeek Team (2024), GPT-4o OpenAI (2024), Gemini-2.5-pro-exp-03-25 DeepMind (2025) **(2) Visual Reasoning Models**: GPT-o3 OpenAI (2025), Claude-3.7-Sonnet-20250219-thinking Anthropic (2025), QVQ-72B QwenTeam (2024).

Table 1: Performance comparison of different participants under 0- to 5-shot settings.

| Model | 0-shot | 1-shot | 2-shot | 3-shot | 4-shot | 5-shot |
|---|---|---|---|---|---|---|
| random-policy | 0.27 | 0.27 | 0.27 | 0.27 | 0.28 | 0.27 |
| human | 0.44 | 0.72 | - | 0.79 | - | 0.79 |
| GPT-o3 | **0.30** | **0.56** | **0.59** | **0.60** | **0.59** | **0.61** |
| GPT-4o | 0.16 | 0.42 | 0.48 | 0.51 | 0.54 | 0.54 |
| Claude-3.7-Sonnet-thinking | 0.19 | 0.39 | 0.43 | 0.44 | 0.46 | 0.47 |
| Claude-3.7-Sonnet | 0.16 | 0.30 | 0.38 | 0.41 | 0.40 | 0.42 |
| QVQ-72B | 0.21 | 0.32 | 0.41 | 0.48 | 0.47 | 0.47 |
| Qwen2.5-VL-72B | 0.14 | 0.26 | 0.28 | 0.32 | 0.33 | 0.37 |
| Gemini-2.5-pro | 0.26 | 0.47 | 0.56 | 0.59 | 0.58 | 0.62 |
| Doubao-1.5-vision-pro | 0.06 | 0.22 | 0.34 | 0.39 | 0.39 | 0.39 |
| DeepSeek-VL2 | 0.19 | 0.24 | 0.33 | 0.42 | 0.42 | 0.42 |

**Requests** All models are configured with default parameters and accessed via their official API.

**Reference Baseline** (1) Human participants are recruited through crowdsourcing platforms and instructed to identify the category of the query image $x_q$ based solely on the provided instruction $\mathcal{I}$ and supporting set $\mathcal{S}_k^n$ as defined in Equation 1, without using any external resources. (2) A random policy is adopted as another baseline, where one category is randomly chosen from the candidate classes for each task. The average performance of this policy is used as the reference x-axis in the following figures.

## 4.2 MAIN EXPERIMENT OF VISUAL FAST MAPPING

The principal findings are presented in Figure 3 and Table 1, highlighting three key observations:

**(1) Enhanced Performance of Mainstream VLMs with Visual Examples.** Contrary to earlier studies reporting negligible improvements, our results demonstrate that mainstream VLMs exhibit clear performance gains when provided with visual examples. This enhancement underscores the evolving capabilities of current VLMs in processing and integrating visual information.

**(2) Persistent Gap Between State-of-the-Art Models and Human Performance.** Despite advancements, even the GPT-o3 (SOTA) fails to reach human-level proficiency, with maximum accuracy $-19.0\%$ and effectiveness $\delta - 16.9\%$. These disparities highlight the superior ability of humans to learn new visual concepts from limited examples.

**(3) Improved Efficiency of Visual Reasoning Models Over Traditional VLMs.** Visual reasoning models (GPT-o3, QVQ-72B, and Claude-3.7-Sonnet-thinking) demonstrate notable improvements in efficiency $\eta + 4.8\%$ and effectiveness $\delta + 4.4\%$, compared to their traditional VLM counterparts (GPT-4o, Qwen-VL-72B, and Claude-3.7-Sonnet). This suggests that models incorporating advanced visual reasoning capabilities are better equipped to handle cross-modal information during inference and that their mechanisms are closer to those of humans.

## 4.3 ANALYSIS ON MECHANISM

A mechanistic analysis is conducted within *VFM Bench* to investigate the underlying strategies for learning from the support set $\mathcal{S}_k^n$. Three metrics inspired by cognitive science to quantify the contribution of different cognitive strategies are shown in Table 2: inter-class discrimination, intra-class similarity, and prior knowledge utilization. To assess individual impact, contrastive experiments involve removing a key element associated with each strategy. A significant performance decline upon the removal indicates its critical role within the overall strategy.

Taken together, these metrics provide insight into the underlying mechanisms of visual concept induction. An ideal model should exhibit a balanced and strategic integration of these cognitive behaviors to achieve robust few-shot learning, just as humans do.

Table 2: Experiment settings for mechanism analysis

| Induction Strategy | Data Construction | Exp denotation | Description | Demo | Metric |
|---|---|---|---|---|---|
| Inter-Class Distinctiveness | $\mathcal{F}_{\mathrm{ndni}}(\mathcal{D})$: Negative Demo Noise | w/ negative noisy demo | Demo images of negative categories (different from the query image) are replaced by pure noise. | | $\phi_{\mathrm{ndni}} = \phi(\mathcal{F}_{\mathrm{ndni}}, \theta, K, \mathcal{D})$ |
| Intra-Class Prototype Formation | $\mathcal{F}_{\mathrm{ichi}}(\mathcal{D})$: Intra-Class Homogeneity | w/ replicate demo | Multiple identical demo images are used within a category, contrasting with the standard setting where demos from the same class exhibit diversity. | | $\phi_{\mathrm{ichi}} = \phi(\mathcal{F}_{\mathrm{ichi}}, \theta, K, \mathcal{D})$ |
| Prior Knowledge Utilization | $\mathcal{F}_{\mathrm{fdli}}(\mathcal{D})$: Demo Label Fabrication | w/ fabricated demo | Real category names are replaced with meaningless fabricated terms. | | $\phi_{\mathrm{fdli}} = \phi(\mathcal{F}_{\mathrm{fdli}}, \theta, K, \mathcal{D})$ |

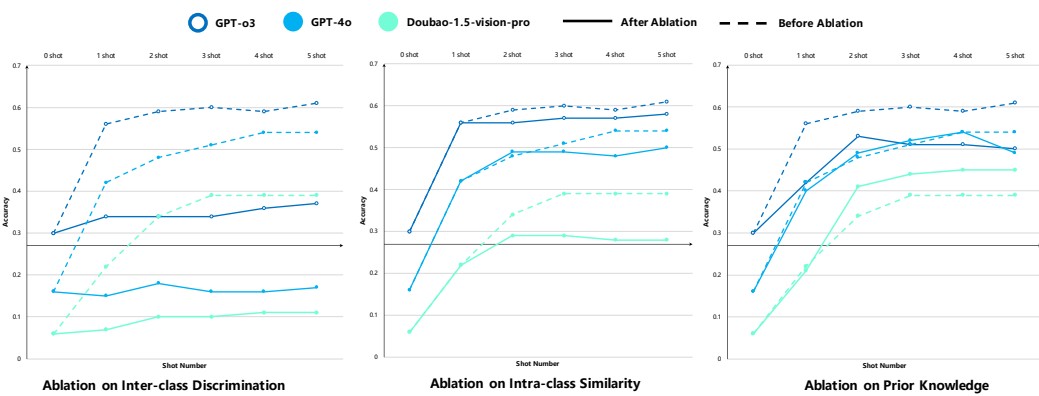

Figure 6: **Ablation experiments for three different inductive strategies**, showing that the model primarily adopts simple contrast among images, rather than utilizing prior language knowledge to perform test-time cross-modal alignment, as humans do.

The results for three typical models are visualized in Figure 6, where the solid lines represent the original experiment and dashed lines denote results after applying the data processing function $\mathcal{F}(\mathcal{D})$. Detailed results for more models are summarized in Table 4.

(1) **Inter-class distinctiveness serves as a key strategy in VLMs.** A sharp drop in the *w/ negative noisy demo* condition highlights the model's reliance on contrast between categories.

(2) **Intra-class similarity is also considered but is of lesser importance.** The performance decline in the *w/ replicate demo* setting is milder, suggesting a secondary role in decision-making.

(3) **Prior knowledge contributes inconsistently.** Results from the *w/ fabricated demo* condition show variation across models; a clear drop can be observed for visual reasoning models, indicating their adoption of prior language knowledge for test-time image-text alignment.

To provide evidence from another perspective, we conducted a domain-specific analysis, as depicted in Figure 7. Our findings reveal that **humans achieve stable performance across domains, while VLMs vary considerably.** VLMs exhibit significant variability, with strengths and weaknesses differing markedly between domains. Humans leverage extensive common-sense knowledge, enabling them to form concepts and make inferences even with limited visual cues. VLMs, however, are constrained by the scope and diversity of their training datasets, lacking the ability to conduct test-time image-text alignment. Although early-stage improvements were observed in the main experiment, current visual reasoning models still struggle to generalize across diverse domains.

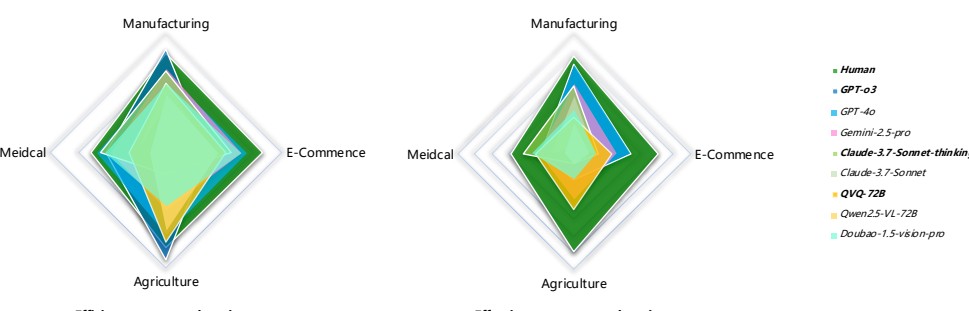

**Efficiency $\eta$ across domains**          **Effectiveness $\delta$ across domains**

Figure 7: **Visualization of Domain-Level Variation**. Compared to the stability of human performance, the greater variation across domains illustrates that VLM performance is constrained by the limited scope of image-text alignment data preventing them from effectively conducting VFM at test time.

Table 3: Experiment settings for cause localization

| Cause Localization | Data Construct | Exp denotation | Description | Demo | Metric |
|---|---|---|---|---|---|
| Fine-grained Detection | $\mathcal{F}_{\mathrm{hkfi}}$: Highlights Key Feature | w/ mask + enhance | Highlights the key feature in the query image by adding a prominent red bounding box around the region. | 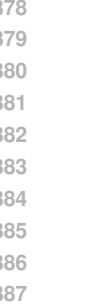 | $\phi_{\mathrm{hkfi}} = \phi(\mathcal{F}_{\mathrm{hkfi}}, \theta, 0, \mathcal{D})$ |
| Alignment Inadequacy | $\mathcal{F}_{\mathrm{vdpi}}(\mathcal{D})$: Visual Demo Provided | w/ $K$ shots | Provides $K$ different visual images for each category using a naive random selection strategy. | | $\phi_{\mathrm{vdpi}} = \phi(\mathcal{F}_{\mathrm{vdpi}}, \theta, K, \mathcal{D})$ |
| Lack of Knowledge | $\mathcal{F}_{\mathrm{dipi}}(\mathcal{D})$: Detailed Instruction Provided | w/ detailed instruction | Provides additional prior knowledge by offering a concise description for each class. | | $\phi_{\mathrm{dipi}} = \phi(\mathcal{F}_{\mathrm{dipi}}, \theta, 0, \mathcal{D})$ |

## 4.4 ANALYSIS ON CAUSE LOCALIZATION

In order to pinpoint the failures in zero-shot classification and illustrate the necessity of providing visual examples, we investigate three potential causes, as illustrated in Figure 11. Detailed experimental settings are described in Table 3.

(1) **Overly Subtle Visual Differences** which are likely lost during the visual encoding process. To mitigate this issue, we enhance the image by introducing visual markings, a technique commonly adopted in fine-grained visual recognition tasks Qian et al. (2015); Tang et al. (2025), to assess whether the failure stems from the model's inability to capture fine-grained visual cues.

(2) **Inadequate Image-text Alignment**. On one hand, the information density mismatch between the visual and linguistic modalities makes it difficult to capture all visual nuances in textual form. On the other hand, query images may belong to long-tail or out-of-distribution objects not well-represented in the training corpus. Both factors contribute to weak cross-modal alignment. Accordingly, we introduce a basic VCFM setting with one example per category to examine whether such a test-time alignment aids in understanding unfamiliar visual concepts.

(3) **Insufficient Domain Knowledge** within the language backbone. The visual concepts in our task are highly domain-specific, and the model may lack the necessary semantic grounding to associate them with the corresponding visual features. We enrich the textual instructions by adding detailed descriptions of each category to test this hypothesis.

As shown in Figure 8 and Table 4, neither the image enhancement nor the addition of detailed textual instructions led to significant performance gains compared to providing visual examples. The enhanced image queries yielded only a moderate improvement, suggesting that the model failures are not caused by an inability to capture fine-grained details. Similarly, enriching prompts with

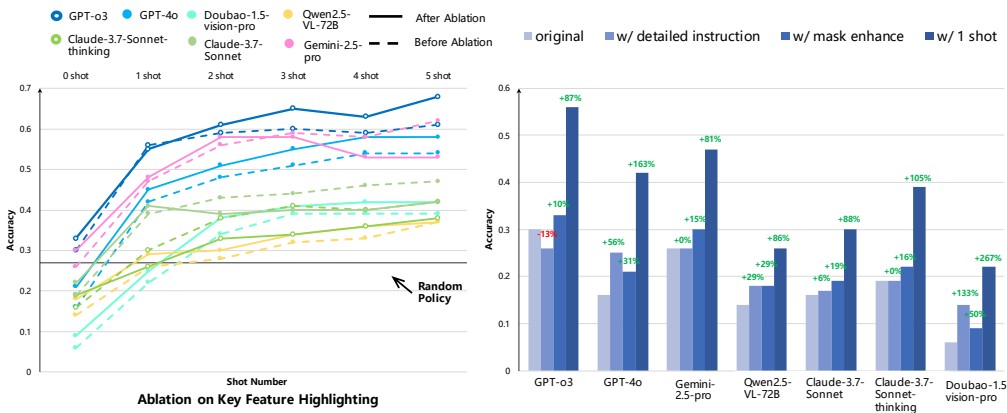

Figure 8: **Ablation experiments for the potential causes of the shortcomings**, The left figure shows that enhancing query images by highlighting anchor features yields minimal accuracy improvement. Conversely, the analysis of different ablation strategies reveals that inadequate image-text alignment is the main factor contributing to the failure of zero-shot classification.

Table 4: Metrics for ablation experiments. Here, **base** denotes the zero-shot classification accuracy for cause localization analysis in Section 4.4, and **ava** represents the average performance across 0 to 5 shots for mechanism analysis in Section 4.3.

| Metrics | base | $\phi_{vdpi}$ | $\phi_{hkfi}$ | $\phi_{dipi}$ | ava | $\phi_{fdli}$ | $\phi_{idhi}$ | $\phi_{ndni}$ |
|---|---|---|---|---|---|---|---|---|
| GPT-o3 | 0.30 | **+87%** | +10% | -13% | 0.54 | -15% | -3% | **-37%** |
| Claude-3.7-Sonnet-thinking | 0.19 | **+105%** | +16% | - | 0.40 | -8% | -17% | **-47%** |
| GPT-4o | 0.16 | **+163%** | +31% | +56% | 0.44 | -2% | -4% | **-63%** |
| Claude-3.7-Sonnet | 0.16 | **+88%** | +19% | +6% | 0.35 | +6% | -24% | **-43%** |
| Gemini-2.5-pro | 0.26 | **+81%** | +15% | - | 0.51 | -9% | -17% | **-37%** |
| Doubao-1.5-vision-pro | 0.06 | **+267%** | 50% | +133% | 0.30 | +13% | -21% | **-69%** |

concept descriptions offered minimal benefit, implying that such knowledge was largely redundant. In contrast, providing a single visual example substantially improves performance, highlighting that the core limitation is insufficient cross-modal alignment from pre-training and underscoring the critical role of test-time image-text alignment.

## 5 CONCLUSION

In this study, we propose a novel benchmark tailored to assess the Visual Fast Mapping (VFM) capabilities of VLMs in industry-relevant classification scenarios. Our findings demonstrate that SOTA VLMs continue to underperform on VFM tasks, particularly when compared to human participants. A deeper analysis reveals that these models primarily rely on inter-class discriminative strategies and remain limited to the data scope of the image-text alignment training stage, with minimal utilization of prior linguistic knowledge available through large-scale language pretraining. Nonetheless, we observe that emerging visual reasoning models begin to exhibit early signs of human-like cognitive processing, indicating the value of test-time cross-modal alignment. These observations highlight the need for next-generation vision systems that can more effectively integrate prior knowledge and demonstrate flexible, human-aligned reasoning capabilities.

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

## A LIMITATIONS

The limitations of our work are summarized as follows:

**Bias and Representativeness:** While the *VFM Bench* encompasses a substantial collection of open-source data across four domains, it does not fully capture the breadth of areas where Visual Fast Mapping (VFM) is pivotal in daily applications, such as long-tail real world objects. Our observations reveal that model performance varies significantly across different tasks, underscoring the necessity for a more diverse and representative benchmark. Future research should focus on expanding the dataset to include a wider array of domains, facilitating a more comprehensive evaluation of Vision-Language Models (VLMs) and their generalization capabilities.

**Deep Analysis within the Benchmark:** To gain deeper insights into model performance, it is essential to analyze fast mapping across varying concept levels. Human learners typically find object-level concepts more accessible than abstract relational concepts. Assigning concept-level labels to each data point can facilitate comparative analyses, as they might have different distributions during image-text alignment training stages. Such analyses can inform future research on the mechanisms underlying vision-language models.

**Consideration of More Complex Tasks:** While classification tasks serve as a foundational assessment of image-text alignment during test time, they represent only the initial stage of cognitive processing. In real-world scenarios, humans not only identify new concepts but also engage in subsequent reasoning, planning, and action. This progression is particularly evident in complex applications such as autonomous driving and embodied AI, where systems must interpret novel, long-tail scenarios daily. In these contexts, Visual Fast Mapping (VFM) capabilities are crucial, enabling models to adapt to unfamiliar situations effectively.

## B AUTHOR STATEMENT AND LICENSE

*VFM Bench* is distributed under CC-BY-NC-SA-4.0. The evaluation code of *VFM Bench* is distributed under the MIT license. We will bear all responsibility for any potential rights violations.

## C DETAILS OF DATA PROCESSING

### C.1 DATA COLLECTION STAGE

As mentioned in Section 3, a total of 31 open-source datasets have been collected, details of which are listed in Table 5.

Moreover, recalling the discussion in Appendix A, four levels of visual concepts are defined in the dataset, as shown in Table 6 denoting a distinct scope of information for different anchor feature, based on which VFM is performing. The instruction for each tasks will follow the definition of certain level.

Finally, detailed statistics of the selected query images are shown in Table 7 with related concept level information.

### C.2 BENCHMARK CONSTRUCTION STAGE

More details about the variants of the formatting and content of Instruction $\mathcal{I}$, Supporting Set $\mathcal{S}$, Query Image $x_q$ and Category Label $y_q$ are illustrated as follows and in Figure 10.

Table 5: Dataset Source

| Domain | Dataset Source | Sample Images | Object Number | Total Categories |
|--------|----------------|---------------|---------------|------------------|
| Manufacturing | VisA (Zhang et al. (2023)) | 10,821 | 12 | 24 |
| | MVTecAD (Bergmann et al. (2021)) | 5,354 | 15 | 88 |
| | MVTec-AD-2 (Heckler-Kram et al. (2025)) | 5,174 | 8 | 16 |
| | MVTec-LOCO (Bergmann et al. (2022)) | 3644 | 5 | 15 |
| | MVTec-3D (Bergmann et al. (2021)) | 4147 | 10 | 50 |
| | ITD (Thomine and Snoussi (2023)) | 5868 | 10 | 20 |
| | ISP-AD (Krassnig and Gruber (2025)) | 559,049 | 3 | 9 |
| | GoodsAD (Zhang et al. (2024)) | 6124 | 6 | 21 |
| | BTAD (Mishra et al. (2021)) | 2,830 | 3 | 6 |
| | NEU-DET (WU (2024)) | 1800 | 1 | 6 |
| E-Commerce | MVTecAD (Bergmann et al. (2021)) | 5,354 | 15 | 88 |
| | MVTec-AD-2 (Heckler-Kram et al. (2025)) | 5,174 | 8 | 16 |
| | WFDD (Chen et al. (2024b)) | 444 | 4 | 11 |
| | AITEX (Silvestre-Blanes et al. (2019)) | 247 | 1 | 2 |
| | Product1M (Zhan et al. (2021)) | 1,182,083 | 1 | 458 |
| | SceneryWatermark(Dmitry (2023)) | 22,783 | 1 | 2 |
| | RiceImage(Cinar and Koklu (2022)) | 75,000 | 1 | 5 |
| | Fabrics(Kampouris et al. (2016)) | 7885 | 1 | 26 |
| | Products-10K (Bai et al. (2020)) | 150,000 | 10,000 | 10,000 |
| | ImageExposures (Cai et al. (2018), Fu et al. (2024)) | 1,000 | 2 | 12 |
| | ClothingAttributesDataset (Chen et al. (2012)) | 1,856 | 11 | 42 |
| Agriculture | Rice-Leaf-Disease(Prajapati et al. (2017)) | 120 | 1 | 3 |
| | TomatoLeaf(kaustubb b (2020)) | 11,000 | 1 | 10 |
| | CAMO (Le et al. (2019)) | 1250 | 1 | 1 |
| | COD10K (Fan et al. (2020)) | 10,000 | 78 | 1 |
| | NC4K (Lyu et al. (2021)) | 4,121 | 1 | 1 |
| | 25-Indian-Bird(Basandrai (2023)) | 22,620 | 1 | 25 |
| | EPLID (Shaheen (2019)) | 3,588 | 1 | 8 |
| | V2-Plant-Seedlings (Giselsson et al. (2017)) | 5,539 | 1 | 12 |
| | MeatDataset(Agistany (2022)) | 365 | 1 | 3 |
| Medical | GMAI-MMBench(Chen et al. (2024c)) | 26,000 | 284 | 78 |
| | MSD Brain(Antonelli et al. (2022), Simpson et al. (2019)) | 750 | 3 | 3 |
| | MSD Spleen (Antonelli et al. (2022), Simpson et al. (2019)) | 61 | 41 | 1 |

Table 6: Definition of four visual concept levels

| Dimension | Definition |
|-----------|------------|
| Object–level | modality-agnostic representation of whole, and focus on nameable entities that remains stable across moderate pose or appearance changes and is directly linkable to lexical tokens |
| Detail–level | identifies those fine-grained, part-scale cues, like feather color, wheel spokes, brand logos, and separates sub-categories inside an object-level class |
| Pattern–level | repeated textures, symmetric motifs or part constellations that are more abstract than details yet below the object level |
| Style-level | global aesthetic or domain statistics, including color palette, lighting regime, painterly brush strokes, that can vary independently of object identity |

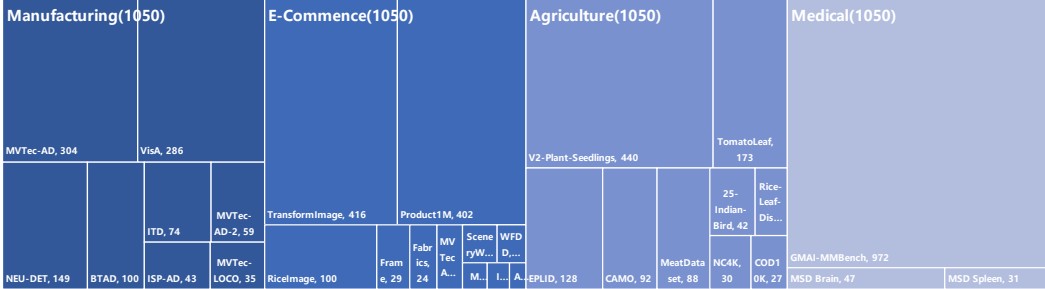

Figure 9: **The raw data distribution of *VFM Bench*** from 31 open-sourced dataset across domains.

Table 7: Detailed Statistics of *VFM Bench*

| Domain | Level | Dataset Source | Sample Images | Object Number | Total Categories | Average Categories |
|---|---|---|---|---|---|---|
| Manufacturing | Object | VisA | 286 | 11 | 22 | 2.00 |
| | Detail | MVTec-AD | 304 | 8 | 48 | 6.37 |
| | Detail | MVTec-AD-2 | 59 | 2 | 4 | 2.00 |
| | Detail | MVTec-LOCO | 35 | 1 | 3 | 3.00 |
| | Detail | ITD | 74 | 9 | 18 | 2.00 |
| | Detail | ISP-AD | 43 | 1 | 3 | 3.00 |
| | Pattern | BTAD | 100 | 3 | 6 | 2.00 |
| | Pattern | NEU-DET | 149 | 1 | 6 | 6.00 |
| E-Commerce | Detail | MVTecAD | 23 | 4 | 25 | 6.09 |
| | Detail | MVTec-AD-2 | 9 | 3 | 6 | 2.00 |
| | Detail | WFDD | 15 | 2 | 4 | 2.00 |
| | Detail | AITEX | 6 | 1 | 2 | 2.00 |
| | Pattern | Product1M | 402 | 6 | 47 | 13.44 |
| | Pattern | SceneryWatermark | 18 | 1 | 2 | 2.00 |
| | Pattern | RiceImage | 100 | 1 | 5 | 5.00 |
| | Pattern | Fabrics | 24 | 1 | 10 | 10.00 |
| | Pattern | Products-10K | 29 | 1 | 2 | 2.00 |
| | Style | ImageExposures | 8 | 1 | 3 | 3.00 |
| | Style | ClothingAttributesDataset | 416 | 1 | 3 | 3.00 |
| Agriculture | Detail | Rice-Leaf-Disease | 30 | 1 | 3 | 3.00 |
| | Detail | TomatoLeaf | 173 | 1 | 5 | 5.00 |
| | Pattern | CAMO | 92 | 1 | 2 | 2.00 |
| | Pattern | COD10K | 27 | 1 | 2 | 2.00 |
| | Pattern | NC4K | 30 | 1 | 2 | 2.00 |
| | Pattern | 25-Indian-Bird | 42 | 3 | 13 | 4.07 |
| | Pattern | EPLID | 128 | 1 | 8 | 8.00 |
| | Pattern | V2-Plant-Seedlings | 440 | 1 | 10 | 10.00 |
| | Pattern | MeatDataset | 88 | 1 | 3 | 3.00 |
| Medical | Detail | GMAI-MMBench | 155 | 21 | 85 | 4.11 |
| | Detail | MSD Brain | 47 | 1 | 3 | 3.00 |
| | Detail | MSD Spleen | 31 | 1 | 3 | 3.00 |
| | Pattern | GMAI-MMBench | 469 | 45 | 153 | 3.91 |
| | Pattern | GMAI-MMBench | 263 | 26 | 95 | 3.94 |
| | Style | GMAI-MMBench | 85 | 8 | 26 | 3.35 |

Table 8: Statistics of *VFM Bench*

| Industry | Dataset Num. | Samples Num. | Task Num. | Concept Num. | Avg. Category per Task |
|---|---|---|---|---|---|
| Manufacturing | 8 | 1050 | 36 | 110 | 3.91 |
| E-Commerce | 11 | 1050 | 22 | 109 | 7.34 |
| Agriculture | 9 | 1050 | 11 | 48 | 6.77 |
| Medical | 3 | 1050 | 102 | 246 | 2.37 |

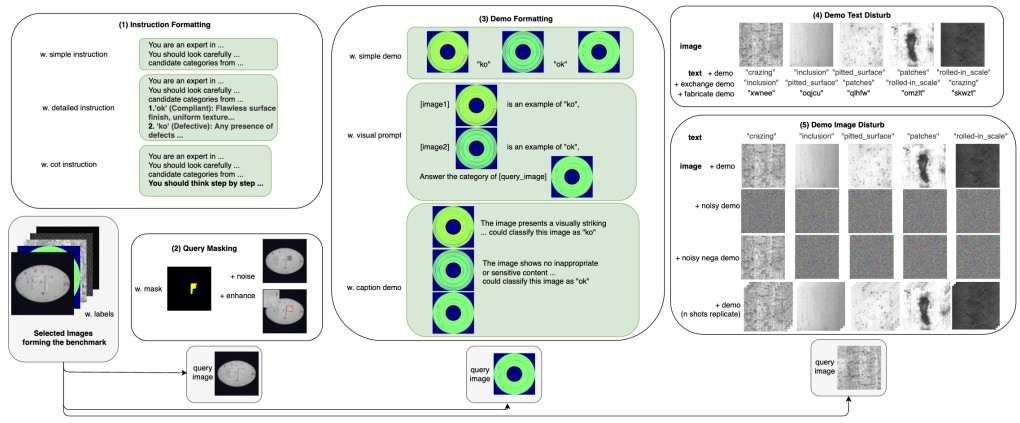

Figure 10: The overview of benchmark construction pipeline, illustrating several variants of formatting and content.

Figure 11: **Three potential causes for the classification failures**, which are mainly caused by visual encoder, cross-modal alignment and pretrained model respectively. Our experiments reveal that the dominant contributor is the inadequacy of cross-modal alignment.

We adopt three types of instructions $\mathcal{I}$:

- **w/ simple instruction**: Follows the Minimal Instruction Principle and presents a basic formulation of the classification task, as introduced in Section 3.2.

- **w/ detailed instruction**: Provides additional prior knowledge by offering concise definitions for each class, making it easier for an average human (and thus the model) to understand the category distinctions.

- **w/ cot instruction**: Encourages step-by-step reasoning to reach the final prediction, following the approach of Wei et al. (2023), considering the recent progress in visual reasoning models.

Moreover, since most fine-grained classification datasets provide ground-truth mask annotations, we construct two additional variants of the query image $\mathcal{X}$ based on the annotated feature regions, in comparison to the original version:

- **w/ enhance**: Highlights the key feature by adding a prominent red bounding box around the masked region. The region is also scaled up and placed in the top-left corner of the image to emphasize the anchor feature.

- **w/ noise**: Removes the anchor feature by replacing the masked region with random noise. The size of the noise patch is set to the average size of the mask regions across all categories in the dataset.

Different variants of the support set $\mathcal{S} = (x_k, y_k)_{k=1}^{n}$, where $y_k \in \mathcal{Y}$, are constructed with various formatting strategies. First, we employ three types of demo formatting for $x_k$ and $y_k$:

- **w/ simple demo**: Follows the standard ICL prompting format where images and corresponding class names are interleaved.

- **w/ caption demo**: Extends the simple demo format by providing a longer textual description that explains the visual features and the reasoning behind the classification decision.

- **w/ visual prompt**: Treats image and text inputs equally by embedding them in a unified token stream, reflecting a more natural multimodal integration, inspired by Zhao et al. (2024a).

Second, we explore two types of textual perturbations based on the simple demo format, still maintaining inter-class consistency:

- **w/ exchange demo**: Category labels are swapped among demos.

- **w/ fabricated demo**: Real category names are replaced with meaningless fabricated terms.

Third, we investigate three types of visual perturbations in the demos, again based on the simple demo format:

- **w/ total noisy demo**: All demo images are replaced with random noise.

- **w/ negative noisy demo**: Only images from negative categories are replaced with noise.

- **w/ replicated demo**: Multiple identical demo images are used within a category, contrasting with the standard setting where demos from the same class exhibit diversity.

## D    CASE STUDY: ATTENTION ON VISUAL TOKENS

In this section, we explain why visual reasoning models could achieve better test-time image-text alignment than traditional VLMs based on a case study, using Attention Rollout for Vision Transformers  Abnar and Zuidema (2020).

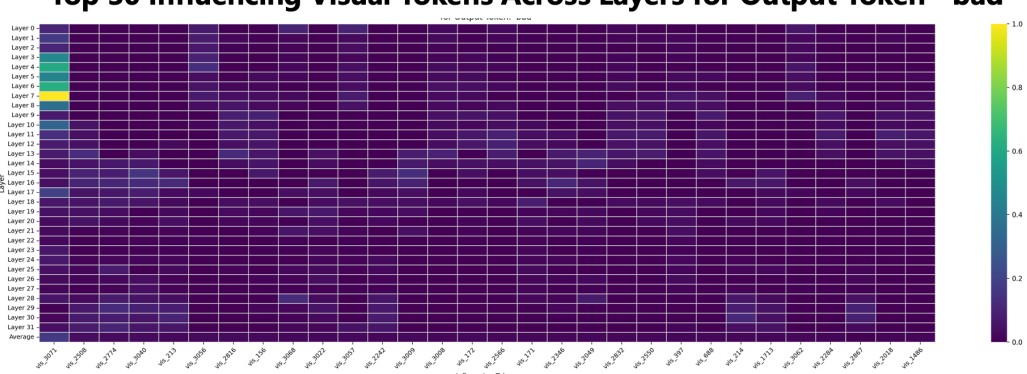

**Top 30 Influencing Tokens Across Layers for Output Token "bad"**

**Top 30 Influencing Visual Tokens Across Layers for Output Token "bad"**

Figure 12: The distribution of attention of the output token for one category (positive case), revealing a clear sparsity in visual token utilization and uneven attention distribution across image sequences.

As shown in Figure 12, test-time attention analysis reveals a clear sparsity in visual token utilization and uneven attention distribution across image sequences. The upper panel illustrates attention values across layers for the top 30 tokens, where only one visual token appears—indicating poor visual-textual alignment. The lower panel highlights the most attended visual tokens, with the query and first demo images dominating around 70% of the attention, echoing prior findings on positional bias toward sequence boundaries. If models could dynamically adjust attention during inference, they may better support efficient Visual Fast Mapping (VFM).

Notably, GPT-o3's multi-modal chain-of-thought behavior exemplifies such adaptive alignment: when uncertain, the model re-inspects ambiguous images or zooms into fine-grained regions by introducing them again in the reference stage. This dynamic cross-modal reasoning offers a promising direction for achieving human-level VFM.

## E    DETAILS OF CROWDSOURCING PARTICIPANTS

Five participants were employed to answer questions in *VFM Bench* on the crowdsourcing platform of Baidu AI Cloud. Their labeling user interface is shown in Figure 13. They were instructed to answer questions solely based on the provided guidelines and examples, without utilizing any external resources. The sequence of data is randomized across different experiments to reduce the impact

of prioritization. Due to cost considerations, we did not conduct human annotation for the 2-shot and 4-shot conditions. We are confident that our current selection of discrete samples is sufficient to capture the essential trends and performance variations of human rapid visual mapping capabilities under few-shot learning. We anticipate that results for 2-shot and 4-shot conditions would show similar patterns to our existing data, and their omission does not fundamentally impact our overall conclusions regarding human performance.

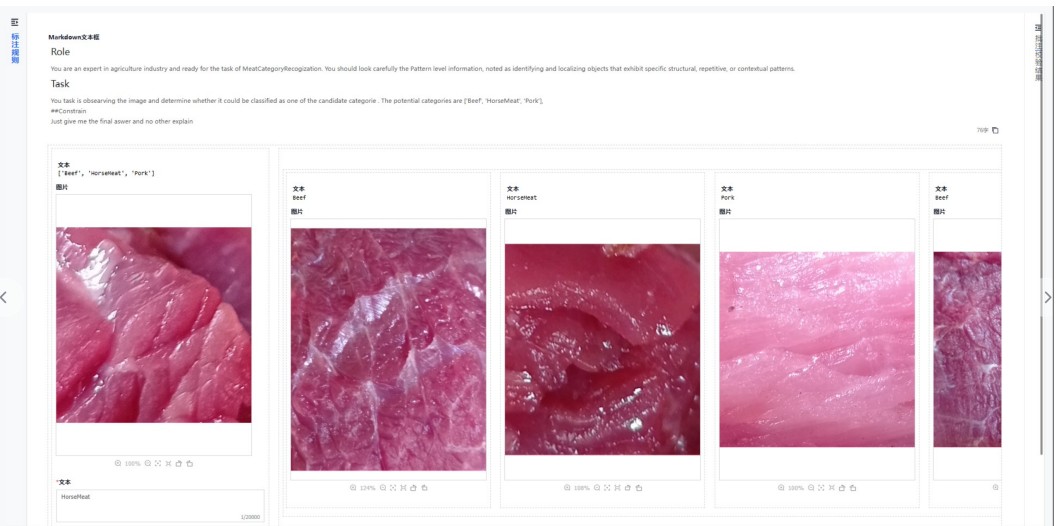

Figure 13: The UI of the crowdsourcing platform to identify the category from instructions and a few visual examples.

## F USE OF LLMS

We promise to use large language models (LLMs) only for checking grammar and spelling errors and ensuring sentences read more smoothly.

