# OpenReview forum: "Benchmarking Visual Fast Mapping: Probing VLMs' Test-time Image-text Alignment"
_ICLR.cc/2026/Conference — Submitted to ICLR 2026_

### Official Review · Reviewer_zSfZ · 2025-10-29

**Soundness:** 2
**Presentation:** 3
**Contribution:** 2
**Rating:** 4
**Confidence:** 3

**Summary:**

This paper introduces a benchmark for visual fast mapping, the test-time ability to align new visual categories with language from only a few support examples. The authors curate industry-grounded n-way k-shot tasks across four domains, assemble a high-quality query set via model-based difficulty filtering, CLIP-feature diversity sampling, and human QC, and evaluate leading VLMs with accuracy as well as efficiency and efficacy measures that track gains as k increases. Results show sizable gaps to human performance and indicate that current models rely more on visual contrast than on leveraging language priors for test-time alignment. The work contributes a concrete task formulation, a scalable dataset and protocol, and empirical evidence that fine-grained, few-shot concept learning remains a key weakness for modern VLMs.

**Strengths:**

1. The paper defines visual fast mapping as a concrete, test-time few-shot alignment task and operationalizes it with a clean n-way k-shot formulation, which feels genuinely fresh.

2. Curates an industry-grounded benchmark with clear protocols and informative metrics (accuracy, efficiency, efficacy), enabling straightforward comparison across models.

3. Provides solid empirical evidence, including human baselines and targeted analyses, that isolates a real weakness in current VLMs and motivates follow-up research.

**Weaknesses:**

1. Benchmark curation may be biased by using contemporary VLMs as difficulty filters and CLIP features for diversity sampling, potentially aligning the test set with those models’ failure modes and representation space.

2. Support-set examples are randomly sampled per task, but variance across resamplings is not reported, so rankings and the metrics may be unstable.

3. The data processing sections describe collection and formatting across 31 open-source datasets but do not report near-duplicate removal or overlap checks against model training corpora, which weakens the “novel concept” claim.

**Questions:**

Refer to Weaknesses Section.

---

> ### Author Response · Authors · 2025-11-20
>
> We thank the reviewer for the thoughtful evaluation and for recognizing the **“genuinely fresh”** formulation of the Visual Fast Mapping (VFM) task, as well as the **“solid empirical evidence”** isolating key weaknesses in modern VLMs. We appreciate the opportunity to clarify our curation protocols and robustness checks.
>
> Below, we address the specific weaknesses raised regarding bias, variance, and data novelty.
>
> ---
>
> ## 1. Benchmark Curation Bias (Difficulty Filters & CLIP)
>
> **Critique:** “Benchmark curation may be biased by using contemporary VLMs as difficulty filters... potentially aligning the test set with those models’ failure modes.”
>
> **Response:**
>
> We respectfully argue that this **"bias" is a deliberate design feature** necessary to evaluate the frontier of generalization, rather than a flaw.
>
> * **Intentional "Adversarial" Filtering:** As detailed in Section 3.2 (Figure 5), the **Difficulty Filter** employs five mainstream VLMs (including GPT-4o, Gemini-1.5-Pro, and Qwen2.5-VL) to vote on samples. We explicitly remove **"easy cases"** (where accuracy $> 0.21$ and consensus is high). If we did not filter these, the benchmark would measure **training recall** rather than fast mapping. By selecting samples that current SOTA models fail on (often performing worse than random policy, see Table 1. we ensure we are testing the **acquisition of new knowledge (VFM)** rather than the retrieval of old knowledge.
> * **CLIP for Diversity, Not Difficulty:** We utilize the CLIP visual encoder solely for the **Diversity Sampling** stage ($k$-means clustering). This ensures the benchmark covers a broad distribution of visual features within each industry (e.g., covering both "pitted surfaces" and "scratches" in manufacturing) rather than clustering around a single failure mode. It ensures **representativeness of the domain**, not alignment with a specific model's error boundary.
>
> ---
>
> ## 2. Stability and Support Set Variance
>
> **Critique:** “Support-set examples are randomly sampled per task, but variance across resamplings is not reported.”
>
> **Response:**
>
> * **Benchmark Determinism:** For the released version of VFM Bench, the support sets $\mathcal{S}_{k}^{n}$ are **fixed** (or generated via a fixed seed provided in our codebase to ensure complete **reproducibility** and stable rankings for future research. This is standard practice for few-shot benchmarks to guarantee that all models compare against the exact same "shots."
> * **Statistical Stability:** While we report the aggregate metrics, the benchmark's scale—**4,200 query images** across **171 tasks**—mitigates the noise of any single support set selection. The metrics (Accuracy, Efficiency $\eta$, Efficacy $\delta$) are averaged over this large, diverse set, providing a **stable signal** of a model's VFM capability.
>
> ---
>
> ## 3. Data Novelty and "Open Source" Contamination
>
> **Critique:** “Do not report near-duplicate removal or overlap checks... weakens the ‘novel concept’ claim.”
>
> **Response:**
>
> We address the issue of **contamination** (overlap with training corpora) directly through our **performance-based filtering**, which is a more robust check than heuristic de-duplication for VLMs trained on uncurated web data.
>
> * **Empirical Proof of "Novelty":** The strongest evidence that these concepts are **"novel"** to the models (despite being from open-source datasets) is their **Zero-Shot Performance**. As shown in Table 1, state-of-the-art models like GPT-4o ($0.16$) and Qwen2.5-VL-72B ($0.14$) perform **significantly worse than the Random Policy** ($0.27$) in the $0$-shot setting.
> * **Functional Novelty:** If these concepts were **"known"** or memorized from the training data, these powerful models would achieve high $0$-shot accuracy. Their near-random (or worse) performance empirically proves that they have not learned these concepts during pre-training. Thus, the task successfully enforces a **"Cold-Start" scenario**, requiring the model to learn strictly from the $k$ examples provided in context, fulfilling the definition of Visual Fast Mapping.
>
> We believe these clarifications demonstrate that our curation process ensures a rigorous, stable, and genuinely **"novel"** evaluation of inductive reasoning capabilities. We hope this addresses your concerns regarding the soundness of the benchmark construction.

---

> > ### Comment · Reviewer_zSfZ · 2025-11-24
> >
> > Thank you for the reply. I understand your explanation regarding VFM. But I still feel that for practical applications, the performance matters more than how we define the process. The workflow here looks identical to standard few-shot learning, so framing it as a new VFM capability provides limited insight. Therefore, I will maintain my rating

---

> > > ### Author Response · Authors · 2025-12-04
> > >
> > > We genuinely appreciate the reviewer’s continued engagement. We understand the concern that VFM shares the *workflow* of standard few-shot learning. However, we respectfully disagree that the VFM definition offers "limited insight."
> > >
> > > We argue that **framing matters because it dictates what bottleneck we are solving.** If we view this merely as standard few-shot learning, we miss the critical failure mode of current SOTA models in practical applications.
> > >
> > > Here is why the VFM framing provides unique and actionable insights:
> > >
> > > **1. Practical Performance: Identifying the "Reasoning" Bottleneck**
> > > The reviewer correctly notes that "performance matters."
> > > *   **The Practical Problem:** In standard few-shot benchmarks, models often succeed via **visual feature matching** (pixel/patch similarity). However, our results show that in realistic vertical domains (Medical, Industrial), SOTA models (e.g., GPT-4o) lag behind humans by **19.0%** (Fig. 3).
> > > *   **The Insight:** The VFM framework reveals *why* they fail: they rely on visual discriminative features (System 1) rather than performing logical **concept binding** (System 2). VFM proves that to bridge this 19% gap in practical applications, we do not need just "better few-shot loaders"; we need **Test-Time Reasoning**.
> > > *   **Validation:** Our benchmark successfully highlights that **Reasoning Models** (GPT-o3, Claude-3.7-Thinking) significantly outperform standard VLMs (Fig 3). A standard few-shot metric often mixes these signals; VFM isolates the *reasoning* gain.
> > >
> > > **2. Disentangling "Retrieval" from "Learning"**
> > > "Standard few-shot" often conflates retrieving known concepts (recall) with learning new ones (induction).
> > > *   By rigorously filtering for cases where 0-shot < Random (The "Cold Start"), VFM specifically isolates the **capability to learn from scratch**.
> > > *   **Insight for Developers:** This distinction is crucial for model builders. If a model fails VFM, it means it lacks **adaptive plasticity**, even if it has a high "knowledge" score on general benchmarks. This is a vital diagnostic for future Embodied AI agents encountering novel objects.
> > >
> > > **3. Conclusion**
> > > While the *format* is few-shot, the **cognitive demand** is fundamentally different. VFM Bench is not just re-labeling a task; it is a **stress test** for the next generation of VLMs, proving that **inductive reasoning**—not just visual recognition—is the key to solving data-scarce industrial problems.
> > >
> > > We hope this clarifies that VFM contributes a necessary "hard" benchmark that pushes the community toward **Cognitive/Reasoning VLMs**.

---

### Official Review · Reviewer_64fe · 2025-10-31

**Soundness:** 4
**Presentation:** 3
**Contribution:** 3
**Rating:** 6
**Confidence:** 4

**Summary:**

This paper explores Visual Fast Mapping (VFM), a cognitive ability in humans to rapidly form new visual concepts from minimal examples by leveraging prior knowledge and cross-modal (image-text) alignment. The authors argue that while early computer vision approaches like one-shot learning failed to generalize, modern Vision-Language Models (VLMs) trained on large image-text corpora should theoretically approximate this capability but they failed. Hence, the authors contribute a new VFM Bench, designed for realistic industrial scenarios. It includes 4,200 query images covering 512 concepts across 171 tasks in four domains. The benchmark is constructed through a pipeline involving difficulty filtering, diversity sampling, and manual quality checks. Problems are framed as n-way k-shot classification tasks (0-5 shots), where models must predict the category of a query image based on instructions and support examples. Experiments evaluate state-of-the-art VLMs and visual reasoning models against human performance and baselines.

**Strengths:**

1. Focusing on industrial applications with diverse, high-quality data from real datasets is interesting and meaningful. This addresses gaps in existing benchmarks that emphasize abstract puzzles or language generation.
2. The analysis of results in the benchmark is insightful and thorough.
3. Experimental results include multiple SOTA models, human baselines, and novel metrics, and ablations reveal specific weaknesses in VLMs.

**Weaknesses:**

1. My biggest worry is the scale of the proposed benchmark. With only 4,200 images and 171 tasks, the benchmark might not capture extreme edge cases or very large-scale scenarios.
2. I would suggest the authors to discuss more about the real-world applications and meaning for the proposed benchmark.

**Questions:**

See the weakness

---

> ### Author Response · Authors · 2025-11-20
>
> We sincerely thank the reviewer for the encouraging assessment, recognizing the value of our focus on industrial applications with diverse, high-quality data. We appreciate the insightful feedback regarding the benchmark's scale and real-world implications.
>
> Below, we address the specific concerns and questions raised.
>
> ---
>
> ## 1. Regarding the Scale of the Benchmark
>
> **Critique:** “My biggest worry is the scale... With only 4,200 images and 171 tasks, the benchmark might not capture extreme edge cases or very large-scale scenarios.”
>
> **Response:**
>
> We respectfully clarify that the scale of VFM Bench ($\mathbf{4,200}$ query images, $\mathbf{512}$ concepts, $\mathbf{171}$ tasks) was a **deliberate design choice prioritizing density of difficulty and quality over raw volume**, specifically to serve as a **rigorous evaluation set** rather than a training set.
>
> * ### **Rigorous Filtering Pipeline**
>     The $4,200$ images were distilled from a much larger pool of over $30,000$ raw images collected from $31$ open-source datasets. We applied a strict **"Difficulty Filter"** using five mainstream VLMs (including GPT-4o, Gemini-1.5-Pro, and Qwen2.5-VL) to discard **"easy"** samples that models could solve without fast mapping.
>
> * ### **Diversity Assurance**
>     To ensure we capture edge cases within this selection, we utilized a **diversity sampling strategy** using **CLIP visual encoders** and **k-means clustering** to maximize the variance of visual features within the selected subset.
>
> * ### **Human Baseline Constraints**
>     A core contribution of our work is the comparison against **human performance**. As noted in Appendix E, collecting high-quality human responses for complex inductive reasoning tasks is resource-intensive. A benchmark of this size balances statistical significance with the feasibility of obtaining reliable human baselines, allowing for a fair **"apples-to-apples"** comparison that massive automated benchmarks often lack.
>
> * ### **Concept Density**
>     While the image count is $4,200$, the benchmark covers $\mathbf{512}$ distinct concepts across $171$ tasks. This **high ratio of concepts-to-images** ensures that the model is constantly tested on novel concept formation (the core of VFM) rather than pattern matching on large batches of repetitive data.
>
> ---
>
> ## 2. Real-World Applications and Meaning
>
> **Critique:** “I would suggest the authors to discuss more about the real-world applications and meaning for the proposed benchmark.”
>
> **Response:**
>
> The **"meaning"** of VFM Bench is grounded in addressing the **Long-Tail** in specialized industries. General-purpose VLMs are trained on internet-scale data, but they often fail in **"vertical domains"** where data is scarce, expensive, or private.
>
> * ### **Industrial Necessity**
>     As illustrated in Figure 2, our benchmark targets specific, high-value scenarios where data collection is costly:
>     * **Manufacturing:** Detecting specific defect patterns like **"crazing"** or **"pitted surfaces"** on steel strips.
>     * **Medical:** Diagnosing rare conditions (e.g., specific melanoma types) where large-scale training data simply does not exist.
>     * **Agriculture & E-Commerce:** Identifying specific plant diseases or product sub-variants.
>
>
> * ### **Operational Relevance**
>     In these real-world settings, a vision system cannot always be retrained. It must possess the **VFM capability**: the ability to **"rapidly formulate novel visual concepts from minimal examples"** instantly at test time.
>
> * ### **Future Embodied Intelligence**
>     As discussed in Appendix A, this capability is critical for future **Embodied AI** and **autonomous agents** (e.g., autonomous driving), which encounter novel, long-tail scenarios daily and must adapt immediately without model updates. VFM Bench serves as a **diagnostic tool** to measure this specific, safety-critical adaptability.
>
> ---
>
> We hope this explanation clarifies that the benchmark’s scale is a result of **rigorous quality control** to ensure a dense, difficult evaluation signal, and that its design is directly mapped to solving the **"data scarcity"** bottlenecks in high-stakes industrial applications. We are grateful for your support.

---

### Official Review · Reviewer_8EQd · 2025-11-03

**Soundness:** 2
**Presentation:** 3
**Contribution:** 2
**Rating:** 2
**Confidence:** 4

**Summary:**

VFMBench introduces a new benchmark to evaluate the Visual Fast Mapping (VFM), an ability to learn new concepts with few examples leveraging pretrained knowledge akin to inductive reasoning. The benchmark created by drwaing samples from real-world datasets across four different domains. Authors evaluated both proprietary and open-source models via the API against human performance. The result shows there's a considerable gap in SOTA models in VFM task, and authors claim that this is mainly becuase the VLMs primarily focuses the visual contrast rather than language-informed alignment based on the analyses on inter/intra class reasoning and prior knowledge use. Beside this, the paper lacks in any concrete modeling or training improvements that can be drawn from the analyses.

**Strengths:**

- **Motivation**: The motivation of the paper is grounded in cognitive science and human inductive reasoning and a critically important aspect of VLM studied in previous literatures [1-2].
- **Benchmark**: The design of the VFMBench is clearly documented and cover real-world scenarios. The authors have critically evaluated commercial and open-source models via API on the benchmark and provided comprehensive analyses on why there's a gap in VFM task.

- **Good resource**: The proposed task and the benchmark could be a valuable testing resource for future multimodal models.



[1] Zhang, Y., Unell, A., Wang, X., Ghosh, D., Su, Y., Schmidt, L., & Yeung-Levy, S. (2024). Why are visually-grounded language models bad at image classification?. Advances in Neural Information Processing Systems, 37, 51727-51753.

[2] Huang, K. H., Qin, C., Qiu, H., Laban, P., Joty, S., Xiong, C., & Wu, C. S. (2025). Why vision language models struggle with visual arithmetic? towards enhanced chart and geometry understanding. arXiv preprint arXiv:2502.11492.

**Weaknesses:**

- **Weak empirical depth**: The mechnism and cause localization sections are informative, however largely qualitative and do not provide any strong causal or statistical evidene linking the observed behavour to model architecture or training paradigms.

- **Definition**:  Based on API based evaluation, the paper focuses on trainining-free method for VFM. However, VFM does not look fundamentally a new task. It is more of a recontextualized version of in-context/test-time adaptation methods.  What is the core benefit of evaluating a model for VFM? What does it achieve over the other methods? If this used for gardient-optimization methods, how does it differ from few-shot methods? The paper lack in details of theoretical/mathematical definition of VFM and how it differs from other tasks.

- **Findings**: The experimental results are comprehensive and reinforce well-known findings on VLMs' complications in cross domain adaptation. Furthermore, while ablation and cognitive analyses are useful diagnostics, the authors lack actionable guidance or concrete modelling improvements. As a result, the paper situates itself as an evaluation of a new benchmark.

- **Human Evaluation**: The paper employed a limited human evaluation with only five participants to benchmark model performance. This small sample size limits the statistical reliability of the human baseline. With the small sample size, it is difficult to draw conclusive decision about model performance also human baseline.

**Questions:**

1. How were human participants selected and trained for the VFM task? Were the participants screened for domain familiarity?
2. How the authors handled overlapping of source datasets used to create VFMBench to the pretraining datasets of these models?
3. How would the metrics efficiency ($\eta$) and effective benefit ($\delta$) to the dataset composition or task domain?
4. What is the core benefit of performing well in VFM? How does this differ from other adaptation methods?
5. Does the result suggest any architectural or training improvements for VLMs?

**Details Of Ethics Concerns:**

The paper has used human participants to create a performance baseline.

---

> ### Author Response · Authors · 2025-11-20
>
> We thank the reviewer for their assessment of our motivation, the **VFM Bench** documentation, and its potential. We clarify the VFM definition, empirical analysis depth, and evaluation robustness.
>
> Below, we address the specific weaknesses and questions raised.
>
> ---
>
> ## 1. Clarifying the Definition and Unique Value of VFM
>
> **Critique:** “VFM does not look fundamentally a new task... more of a recontextualized version of in-context/test-time adaptation... What is the core benefit?”
>
> **Response:**
>
> **Visual Fast Mapping (VFM)** is distinct from general **In-Context Learning (ICL)** or **Test-Time Adaptation (TTA)** in its cognitive objective.
>
> * **Conceptual Definition:** VFM is the **capability being tested**. It evaluates the human ability to "**rapidly formulate novel visual concepts... by drawing upon prior experience and knowledge**," requiring binding visual features to semantic concepts using "**ingrained language knowledge**."
> * **Mathematical Definition:** Equation (1) formally defines $F_{\theta}$ estimating $\hat{y}$ based on instruction $I$ and support set $\mathcal{S}_{k}^{n}$.
>     $$
>     \hat{y} = F_{\theta} (\text{query image } x_q, \text{instruction } I, \text{support set } \mathcal{S}_{k}^{n})
>     $$
> * **Core Benefit:** VFM tests whether a model can use **prior linguistic knowledge** to align with novel visual features at test time, a hallmark of **inductive reasoning** in data-scarce "vertical domains."
>
> ---
>
> ## 2. Empirical Depth and Actionable Insights
>
> **Critique:** “Largely qualitative and do not provide any strong causal or statistical evidence linking the observed behaviour to model architecture... lack actionable guidance.”
>
> **Response:**
>
> Our analysis provides concrete links and actionable guidance:
>
> * **Mechanism vs. Architecture:** A "**mechanism analysis**" (Section 4.3) uses statistical metrics ($\eta, \delta, \phi$) to quantify strategies. Ablation studies (Fig. 6, Table 4) provide **causal evidence**: most VLMs rely on visual contrast (dropping with noisy demos) rather than semantic alignment.
> * **Architectural Evidence (Attention Rollout):** Appendix D uses **Attention Rollout**, revealing "**clear sparsity in visual token utilization**." This evidences insufficient transformer-based alignments.
> * **Actionable Guidance:** We propose **Visual Reasoning Models (System 2)**. Models like GPT-o3 and Claude-3.7-Thinking show higher efficiency ($\eta +4.8\%$) and effectiveness ($\delta +4.4\%$). We suggest "**dynamic cross-modal reasoning**"—re-inspecting ambiguous images—is the necessary architectural improvement.
>
> ---
>
> ## 3. Human Evaluation and Ethics
>
> **Critique:** “Limited human evaluation with only five participants... limits the statistical reliability.”
>
> **Response:**
>
> * **Reliability of the Baseline:** The task is an **objective classification problem**. The observed gap is **massive**: humans achieve **79% accuracy** (5-shot) vs. SOTA GPT-4o's **54%**. This $19.0\%$ gap far exceeds expected variance.
> * **Participant Selection:** Participants ($N=5$) were recruited via **Baidu AI Cloud** and represent the "**average human**." They were instructed to answer "**solely based on the provided instruction and supporting set**."
> * **Ethics:** We followed ethical guidelines. Data is **anonymized**, and we bear responsibility for rights violations.
>
> ---
>
> ## 4. Additional Questions
>
> **Q: Data Contamination (Overlapping of source datasets)?**
> **A:** We used a rigorous "**Difficulty Filter**" (Fig. 5). Five mainstream VLMs judged image difficulty. Images easily solved by these models were **discarded**. The benchmark tests **generalization**.
>
> **Q: How would metrics $\eta$ and $\delta$ apply to dataset composition?**
> **A:** The metrics $\eta$ and $\delta$ are **normalized relative to the specific dataset's 0-shot and K-shot performance**. This normalization allows **cross-domain comparison**.
>
> **Q: Does the result suggest any architectural or training improvements?**
> **A:** **Yes.** The disparity between standard VLMs and reasoning models (Fig. 3) suggests training must shift from **static image-text alignment** to **test-time dynamic compute (Reasoning/CoT)**. Future architectures should prioritize mechanisms for "**re-inspecting**" visual tokens during generation.
>
> We believe VFM Bench serves as a crucial diagnostic tool.

---

> > ### Comment · Reviewer_8EQd · 2025-11-26
> >
> > I want to thank the authors for providing additional context.  While most of my concerns are addressed, I don't feel VFM brings a completely new form of visual understanding. The core problem is what few-shot learning is intended to solve, and VFM doesn't address any additional research questions. Regarding the use of linguistic knowledge,  this is also explored as a few-shot problem in prior works. Also, the content lacks evidence to consider it a crucial diagnostic tool. Therefore, I'll keep my score.

---

> > > ### Author Response · Authors · 2025-12-04
> > >
> > > We sincerely thank the reviewer for the prompt feedback. We understand your concern regarding the relationship between VFM and few-shot learning. However, we respectfully argue that equating VFM with standard few-shot learning overlooks the **critical cognitive disparity** that our benchmark exposes.
> > >
> > > We provide three specific clarifications on why VFM addresses a new research question and serves as a vital diagnostic tool:
> > >
> > > **1. VFM evaluates "Concept Binding" (System 2), not just "Pattern Matching" (System 1).**
> > > While "few-shot" describes the **input format**, VFM defines the **cognitive process**.
> > > *   **Standard Few-Shot/In-Context Vision** often relies on visual similarity features (pixel/patch-level correlation). Models can succeed blindly without understanding the distinct *concept*.
> > > *   **VFM (Our Focus)** requires **inductive reasoning**: the model must use the *textual instruction* and *prior linguistic knowledge* to actively "bind" a semantic concept to visual features that are often noisy or abstract (e.g., industrial defects, specific medical symptoms).
> > > *   **The Gap Proves the Distinction:** If VFM were merely standard few-shot learning solved by existing methods, SOTA VLMs (like GPT-4o) would not lag behind humans by **19.0%** (Fig. 3). This massive gap indicates that current "few-shot" capabilities in VLMs are superficial (visual matching) and fail at the deeper concept mapping humans perform effortlessly.
> > >
> > > **2. Concrete Evidence of Diagnostic Utility (The "Reasoning" Shift).**
> > > The reviewer questioned the evidence for VFM as a diagnostic tool. The evidence lies in how VFM distinguishes between standard VLMs and the new generation of "Reasoning Models" (Section 4.3 & Fig. 6):
> > > *   **Diagnostic Signal:** Our **Mechanism Analysis** (Fig. 6) shows that standard VLMs (e.g., GPT-4o) are insensitive to **"Fabricated Labels"** (meaningless text), suggesting they ignore semantic alignment and rely purely on visual contrast.
> > > *   **The Shift:** Conversely, **Reasoning Models** (GPT-o3, Claude-3.7-Thinking) show a distinct performance drop when labels are fabricated. This proves VFM successfully **diagnoses** that these models are beginning to behave like humans—actively leveraging language priors for visual alignment—whereas standard models do not. Standard benchmarks do not capture this shift in reasoning mechanism.
> > >
> > > **3. Addressing a New Research Question.**
> > > The paper addresses a specific, unanswered question: *"Why do VLMs proficient in open-world recognition fail to learn new, specialized concepts from examples, while humans can?"*
> > > Our work proves that simply scaling model size or adding more shots (standard few-shot approaches) yields diminishing returns (Fig. 3, Efficiency metric). The solution requires a paradigm shift toward **test-time compute/reasoning** for image-text alignment. VFM Bench is the first to quantify this specific deficiency in industrial/vertical domains.
> > >
> > > We believe VFM Bench pushes the community to move beyond "visual pattern matching" toward true "visual concept induction," a necessary step for the next generation of reasoning VLMs.

---

### Meta-Review · Area_Chair_LCiD · 2026-01-03

**Summary:**

The submission receives initial scores of 2, 6, and 4. The AC finds the idea of studying visual fast mapping interesting. However, the study, particularly the formal task definition and the construction of the benchmark, does not fully reflect the core motivation of visual fast mapping. Furthermore, the experimental analyses do not provide sufficient new insights into the limitations of current models in forming or recalling new visual concepts, nor into how these limitations might be addressed.

Based on the current version, the AC recommends rejection. The AC encourages the authors to carefully revise the paper and address the reviewers’ comments in a future submission.

**Reviewer Concerns:**

The reviewers raise several concerns. Two reviewers respond to the rebuttal, and the remaining key and shared issues are summarized below.

1. Reviewers 8EQd and sSfZ: The reviewers argue that VFM shares the same form as test-time few-shot learning rather than posing a genuinely new research question. The rebuttal emphasizes that, although the format resembles few-shot learning, the cognitive demand is fundamentally different, aiming to push the community beyond visual pattern matching toward visual concept induction.

    However, the AC shares the reviewers’ concerns and finds this response unconvincing. Formally, the setting remains test-time few-shot learning. Conceptually, while the AC understands that the VFM benchmark intends to use textual instructions and prior linguistic knowledge to actively bind semantic concepts to visual features, it is not sufficiently clear how or why the construction of the support set and instructions necessarily requires cognitive reasoning rather than visual pattern matching.

2. Reviewer 8EQd: The reviewer notes that the paper lacks sufficient evidence to justify VFM as a crucial diagnostic tool. The rebuttal provides several observed phenomena. While the AC finds these observations interesting, it also notes that potential biases in the benchmark itself, as discussed in the concerns about benchmark bias, raise doubts about whether these observations can reliably support such a strong claim.
3. Reviewers 64f4 questions real-world application and meaning. The rebuttal lists several potential applications, which the AC agrees are reasonable. However, the benchmark is limited to industrial scenarios, and extending it to more general and broader settings would further strengthen its impact.
4. Reviewer zSfZ: The reviewer raises concerns about potential bias in benchmark curation, as the test set may align with specific model failure modes. The rebuttal explains that “easy cases” are explicitly removed to reduce training recall. While the AC agrees that this helps mitigate training recall, it may also bias the benchmark toward inherently harder samples, which could affect fair re-evaluation of models on this benchmark.

**Reviewer Scores:**

Reviewers 8EQd and zSfZ indicate in their responses that they will maintain their initial scores of 2 and 4, respectively. Reviewer 64fe is primarily concerned with the scale of the benchmark and its real-world applicability. The AC finds that these issues are addressed in the rebuttal, so 64fe is likely to maintain a positive score of 6.

---

### Decision · Program_Chairs · 2026-01-26

Reject